# In-Silico Characterization of Estrogen Reactivating β-Glucuronidase Enzyme in GIT Associated Microbiota of Normal Human and Breast Cancer Patients

**DOI:** 10.3390/genes13091545

**Published:** 2022-08-27

**Authors:** Fatima Muccee, Shakira Ghazanfar, Wajya Ajmal, Majid Al-Zahrani

**Affiliations:** 1School of Biochemistry and Biotechnology, University of Punjab, Lahore 52254, Pakistan; 2National Institute for Genomics Advanced Biotechnology (NIGAB), National Agricultural Research Centre (NARC), Islamabad 45500, Pakistan; 3Biological Science Department, College of Science and Art, King Abdulaziz University, Rabigh 25724, Saudi Arabia

**Keywords:** breast cancer, estrogen, probiotics, *Lactobacillus*, reactivation, deconjugation

## Abstract

Estrogen circulating in blood has been proved to be a strong biomarker for breast cancer. A β-glucuronidase enzyme (GUS) from human gastrointestinal tract (GIT) microbiota including probiotics has significant involvement in enhancing the estrogen concentration in blood through deconjugation of glucuronidated estrogens. The present project has been designed to explore GIT microbiome-encoded GUS enzymes (GUSOME) repertoire in normal human and breast cancer patients. For this purpose, a total of nineteen GUS enzymes from human GIT microbes, i.e., seven from healthy and twelve from breast cancer patients have been focused on. Protein sequences of enzymes retrieved from UniProt database were subjected to ProtParam, CELLO2GO, SOPMA (secondary structure prediction method), PDBsum (Protein Database summaries), PHYRE2 (Protein Homology/AnalogY Recognition Engine), SAVES v6.0 (Structure Validation Server), MEME version 5.4.1 (Multiple Em for Motif Elicitation), Caver Web server v 1.1, Interproscan and Predicted Antigenic Peptides tool. Analysis revealed the number of amino acids, isoelectric point, extinction coefficient, instability index and aliphatic index of GUS enzymes in the range of 586–795, 4.91–8.92, 89,980–155,075, 25.88–40.93 and 71.01–88.10, respectively. Sub-cellular localization of enzyme was restricted to cytoplasm and inner-membrane in case of breast cancer patients’ bacteria as compared to periplasmic space, outer membrane and extracellular space in normal GIT bacteria. The 2-D structure analysis showed α helix, extended strand, β turn and random coil in the range of 27.42–22.66%, 22.04–25.91%, 5.39–8.30% and 41.75–47.70%, respectively. The druggability score was found to be 0.05–0.45 and 0.06–0.80 in normal and breast cancer patients GIT, respectively. The radius, length and curvature of catalytic sites were observed to be 1.1–2.8 Å, 1.4–15.9 Å and 0.65–1.4, respectively. Ten conserved protein motifs with *p* < 0.05 and width 25–50 were found. Antigenic propensity-associated sequences were 20–29. Present study findings hint about the use of the bacterial GUS enzymes against breast cancer tumors after modifications via site-directed mutagenesis of catalytic sites involved in the activation of estrogens and through destabilization of these enzymes.

## 1. Introduction

A variety of microbes exist in human gastrointestinal tract (GIT) which may have positive or negative impact on human body. Microbes with good or positive impact are usually known as probiotics. Probiotics are the living microorganisms which confer benefits to human health when administered in the body in sufficient concentrations [1]. They are available as live microbial feed supplements [2]. They exert positive effects on human health such as digestion of lactose, normalization of small bowel-associated microbes, conferring resistance to enteric pathogens, immune system regulation, anticancer by detoxification of carcinogenic metabolites, intestinal wall synthesis, short chain fatty acids and vitamin B and K synthesis, colonization resistance, antihypertensive action, reduction in detoxification, inhibition of *H. pylori*, stimulation of immune response against viruses and production of cofactors and vitamins etc. [3,4]. Generally, probiotics also play a positive role in cancer prevention due to anti-proliferative and apoptotic activities. These cancers include colorectal cancer, gastric cancer, bladder cancer, prostate cancer, breast cancer, skin cancer and leukemia. Incidence of breast cancer has been reported to be reduced in females with post puberty consumption of *Lactobacillus* [4]. *Bacillus subtilis* has also been found to produce a surfactant with anti-tumor effect with respect to breast cancer [5]. In addition to these beneficial microbes, some bacteria of GIT are also harmful to the human body in various ways. For example, some GIT microbiota enzymes have been found to enhance the estrogen level in blood circulation leading to breast cancer incidence which constitutes 30% of female cancers [6,7].

Due to health benefits, probiotics contribute to the development of commercially important functional foods and pharmaceutical formulations. For example, *L. rhamnosus*, *L. acidophilus*, *L. casei*, *L. plantarum* and *Bifidobacterium* species are commonly used in dairy food production including fermented milk, yogurt, cheese, and baby milk powder [8]. Probiotics including *L. Rosell* 52, *Bifidobacterium*, *S. cerevisiae*, *L. salivarius*, *L. reuteri* and *L. acidophilus* NCFB 1748 have potential applications in pharmacy due to their association with prevention of pathogens, travelers diarrhea and antibiotic associated diarrhea, reduction in upper respiratory tract infections and constipation improvement etc. [9,10].

In 2014, microbiota presence was verified for the first time in breast tissue [11]. Literature reports presence of unique microbiota in tumor breast tissue as compared to adjacent non-tumor tissue in breast cancer patient. Even the bacterial composition of tumor breast tissue is different from breast tissues of healthy woman [12]. Human mammary and GIT microbiome composition has been reported to show variation under normal and breast cancer states. Several studies reporting breast cancer dysbiosis are present in literature [13,14,15,16]. Both the GIT and mammary β-glucuronidase (GUS) enzyme exhibit high diversification with respect to structure, function, size, sub-cellular localization, substrate specificity and binding and biocatalytic activity. This diversity is contributed by environmental factors including pathological conditions, antibiotics and diet composition [17]. Initiation and progression of breast cancer has a strong association with microbial dysbiosis of breast tissue [12]. A comparison of healthy and breast cancer patient’s microbiota has been explored in human females. Under normal conditions, bacteria belonging to *Roseburi* species are reported to constitute 7% of healthy human microbiota [18]. Lactic acid bacteria possess anticancer potential. A number of these bacteria have been reported to be reduced in breast cancer patients [19]. *Proprionibacterium acnes*, *Coprococcus* spp., *Faecalibacterium prausnitzii*, *Neisseria elongata*, *Propioni acnes*, *Allisonella* sp., *Megasphaera* sp., *Pedicoccus* sp., *Abiotrophia* sp., *Clostridium sensu* and *Variovorax* sp. W03 have also been found in non-tumor tissues [20,21,22]. Bacteria *Bacillus*, *Enterobacteriaceae*, *Staphylococcus* and *Escherichia coli* have been reported in high abundance in breast cancer patients. Histone-2AX (H2AX) phosphorylation assay performed on HeLa cells showed the role of these bacteria in DNA double stranded breaks [19]. A study initiated to analyze breast cancer-associated dysbiosis in pre-menopausal women using targeted metabolomics, 16S rRNA sequencing and cell culture methods confirmed the presence of *Fusobacterium nucleatum*, *Pediococcus*, *Salmonella enterica*, *Corynbacterium*, *Shewanella putrefaciens*, *Enterococcus gallinarum* and *Thermus scotoductus* and *Desulfovibrio* [22]. *Ralstonia* genus has been found in breast cancer tissue through analysis of variable regions of 16S rRNA [23].

β-glucuronidase (GUS) is a glycosyl hydrolase enzyme which converts glycosides into aglycones by hydrolyzing O- or S-glycosidic moieties [24]. In bacteria, this enzyme is encoded by uidA gene. GIT microbiome-encoded GUS enzymes also known as GUSOME exhibit high levels of diversity. Approximately 279 GUS enzymes have been reported in Human Microbiome Project database [25]. The enzymes can be classified into seven groups i.e., Loop 1 (L1), Mini-Loop 1 (mL1), Loop2 (L2), Mini-Loop 2 (mL2), Mini-Loop 1, 2 (mL1, 2), No Loop (NL) and no coverage groups [25,26]. Three types of GUS i.e., BuGUS-1, BuGUS-2 and BuGUS-3 have been reported in *Bacteroides uniformis* [27].

In 2019, the role of β-glucuronidase in reactivation of estrogen was proved experimentally [28]. Estrogen is found in two circulating forms i.e., estradiol and estrone in pre- and postmenopausal women, respectively [29]. During estrogen metabolism, both these forms are conjugated with glucuronic acid in the presence of UDP-glucuronosyl transferase enzymes (UGTs) leading to the formation of estrone 3-glucuronide and estradiol-17-glucuronide [30]. Due to high polarity and hydrophilicity, glucuronidated estrogens have more tendency to dissolve in blood and excrete via urine. However, a major proportion of conjugated forms enters the GI tract via bile and metabolized further [31]. Once in the intestine, glucuronidated estrogens are deconjugated in the presence of GUSOME into aglycones estrone and aglycones estradiol. The activated estrogen is absorbed in mucosa and re-enters blood circulation via a portal vein thus contributing to breast cancer (Figure 1) [32].

Breast cancer has been found to be reduced in human females on stopping estrogen replacement therapy (ERT) [33]. Estrogen has been reported to be a potential biomarker for breast cancer [34]. This is due to its contribution to enhanced proliferation of cancerous cells, angiogenesis and metastasis stimulation and resistance to chemotherapy [35,36,37,38]. Keeping in view the association of estrogen with breast cancer and the significant role of microbial GUS enzyme in reactivation of this hormone, we designed the present study. This study targets the GUS enzymes in bacteria inhabiting GIT of normal and breast cancer patients. Characterization of the GUS enzyme might help us in the manipulation of GIT-associated bacteria including probiotics to reduce estrogen-related cancer risk. Manipulation can be performed to reduce the stability of enzyme, to alter the 3-D configuration and catalytic site of enzymes, thus reducing the breast cancer risk associated with the activity of enzymes.

## 2. Methodology

### 2.1. Protein Sequences

To retrieve protein sequences of bacteria documented in present project, Uniprot database (https://www.uniprot.org, accessed on 21–24 July 2022) was explored [39]. Sequences retrieved, their accession numbers and bacterial species selected for this study are mentioned (Appendix A).

### 2.2. Phylogeny Analysis

To construct the phylogenetic tree initially, protein sequences of nineteen bacteria documented in present study were aligned using Clustal Omega Multiple Sequence Alignment Tool [40]. The aligned file was then subjected to MEGA version 7 [41]. The evolutionary history was inferred using the Neighbor-Joining method [42]. The evolutionary distances were computed using the Poisson correction method and are in the units of the number of amino acid substitutions per site [43]. All positions containing gaps and missing data were eliminated. There were a total of 586 positions in the final dataset.

### 2.3. Prediction of Physicochemical Properties

To explore the physicochemical properties of bacteria, ProtParam tool (https://web.expasy.org/protparam/, accessed on 24 July 2022) was employed. Computed attributes of bacterial proteins include molecular weight, theoretical isoelectric point (pI), half-life, instability index and aliphatic index.

### 2.4. Sub-Cellular Localization and Ontology Analysis

To predict the sub-cellular localization and ontology of uidA encoded GUS protein in the bacteria addressed in present study, CELLO2GO tool (cello.life.nctu.edu.tw/cello2go/, accessed on 25 July 2022) was employed. 

### 2.5. 2D Structure

For secondary structure prediction, SOPMA tool from Network Protein Sequence Analysis (https://npsa-prabi.ibcp.fr/cgi-bin/npsa_automat.pI?page=npsa_sopma.html, accessed on 30 July 2022,) was employed [44]. To predict secondary motif map PDBsum tool (http://www.ebi.ac.uk/thornton-srv/databases/cgibin/pdbsum/GetPage.pl?pdbcode=index.html, accessed on 30 July 2022) was used. To predict the catalytic site of GUS enzyme Caver Web server v 1.1 (https://loschmidt.chemi.muni.cz/caverweb/, accessed on 4 August 2022) was used [45]. The 25 to 75% of protein is made up of secondary structure building blocks [46]. α helix, extended strand, β turn and bends are the basic elements of secondary conformation. It is important to analyze the impact of SNPs on these elements in order to gain an insight into the deleteriousness of SNPs. 

### 2.6. 3D Structure

The three dimensional structures of GUS enzyme for all the microbes were explored using Phyre2 tool (www.sbg.bio.ic.ac.uk/~phyre2/html/page.cgi?id=index, accessed on 26–28 July 2022). The 3D models were visualized using PyMOL. To measure the accuracy of protein model and predict stereochemical characteristics of protein structures, Ramachandran plot was used [47]. To generate Ramachandran plot, SAVES v6.0 (https://saves.mbi.ucla.edu/, accessed on 30 July 2022) was used. Structure quality has been estimated on the basis of peptide bond planarity, hydrogen bonds energy backbone phi and psi angles. Analysis is based on 118 structures of resolution of at least 2.0 angstrom (Å) and R-factor no greater than 20%. 

### 2.7. Conserved Protein Motifs Analysis 

To predict the conserved motifs in protein sequences of probiotics, MEME version 5.4.1 (http://meme.sdsc.edu/meme/meme.html, accessed on 26 July 2022) was used. This tool usually finds three motifs by default however, in the present study we tried to find up to ten motifs. All other parameters were set according to default settings. To estimate the ontology of each individual conserved domain, Interproscan (http://www.ebi.ac.uk/interpro/search/sequence/, accessed on 4 August 2022) was employed.

### 2.8. Predicted Antigenic Peptides Tool

To predict the antigenic determinants of GUS enzymes of all the bacteria included in study, Kolaskar and Tongaonkar method i.e., Predicted Antigenic Peptides Tool was used (https://imed.med.ucm.es/Tools/antigenic.pl, accessed on 26 August 2022). This prediction algorithm depends on amino acids occurrence in experimentally determined epitopes. 

## 3. Results

### 3.1. Phylogeny Prediction Based on uidA Gene Sequence

As we are trying to explore the GUS enzyme among the bacteria inhabiting the GIT of normal and breast cancer patients. Therefore, it is important to gain insight into the evolution of the GUS-coding uidA gene of these bacteria. For this purpose, the phylogenetic tree has been constructed using GUS enzymes sequences (Figure 2). According to this phylogeny study, *F.*
*prausnitzii* 1 and *S. suis* are sharing the same clade so are closely related to each other. *E. coli* strain K12 and *S. enterica* are also originating from the same branch point. *M. bacterium* is not sharing closeness with any of the bacteria as it is not sharing clade. *S. xylosus* and *S. caeli* are closely related with each other and also shared clade with *S. hemolyticus*. These three bacteria are also related with *R. intestinalis*. *P. acnes* and *E. gallinarum* 2 are sharing closeness. *S. aquatilis*, *F. prausnitzii* 3 and *C. amalonaticus* are also related more with each other as compared to other bacteria. *C. comes* and *Bacillus* sp. are showing the common ancestry due to origin from common branch point. *L. rhamnosus* and *F. prausnitzii* 2 are sharing clade with each other. *E. gallinarum* 1 is also originating from the same branch point as that of *L. rhamnosus* and *F. prausnitzii* 2. 

### 3.2. Physicochemical Attributes

The GUS enzymes in most of the microbes comprise of different numbers of amino acids (Table 1). The highest number is 795 and the lowest is 586 in *C. comes* and *M. bacterium*, respectively. The half-life was observed to be the same in all the bacteria i.e., 30 h. The highest isoelectric point (8.92) was observed in the case of *S. aquatilis* NBRC 16722 while the lowest (4.91) was found in *S. caeli*. Extinction coefficient, instability index and aliphatic index also showed variability. Highest (155075) and the lowest values (89980) of extinction coefficient have been observed in cases of *E. gallinarum* 1 and *S. xylosus*, respectively. The instability index is observed to be highest (40.93) in *C. amalonaticus* and lowest (25.88) in *E. gallinarum* 1. As far as the aliphatic index is concerned, the largest value 88.10 was found in the case of *S. suis* and the smallest 71.01 in *L. rhamnosus*.

### 3.3. Sub-Cellular Localization

In normal tissue-associated bacteria, GUS was found to be present in cytoplasm (Table 2, Figure 3). Meanwhile, in *C. comes*, *L. rhamnosus, F. prausnitzii* 1 and *F. prausnitzii* 2, protein was additionally localized in extracellular, outer-membrane, periplasmic space and inner-membrane. The protein in majority of the breast cancer-associated bacteria was found to be localized in cytoplasm with the exception of *M. bacterium* and *S. aquatilis* NBRC 16722 in which it was additionally localized in inner-membrane.

### 3.4. 2D Structure Prediction

Secondary structure composition analysis based on SOPMA tool revealed that α helices, extended strand, β turn and random coil of GUS proteins in bacteria were comprised of amino acids in the range of 22.66 to 27.81%, 22.04 to 25.91%, 5.39 to 8.30% and 41.75 to 47.70%, respectively (Table 3). Secondary Motif Maps predicted using PDBsum tool are shown in Appendix A. The catalytic site properties were predicted using Caver Web. The catalytic sites with most reliable starting points and 100% relative scores with tunnel druggability, bottleneck radius, length and curvature are shown in Figure 4 and Table 4. Druggability, bottle neck radius, length and curvature of predicted catalytic sites were found to be in the range of 0.052 to 0.80, 1.1–2.8 Å, 1.4–15.9 Å and 0.65–1.4, respectively, for GUS enzymes. Highest and lowest tunnel bottleneck radii were observed in *E. gallinarum* 2 and *S. aquatilis*, respectively. Highest and lowest tunnel lengths were found in cases of *E. gallinarum* and *L. rhamnosus*, respectively. Highest and lowest tunnel curvature was observed in *F. prausnitzii* and *P. acnes* and *S. aquatilis*, respectively.

### 3.5. 3D Structure Prediction

Three dimensional configurations of GUS enzymes were obtained through PHYRE2 tool and visualized by PyMol (Figure 5). According to verification by Ramachandran plot, values of quality model were found to be closer to 90% in the most favored region which reflects the accuracy of GUS protein structures in the case of all bacteria (Appendix A, Table 5). 

### 3.6. Conserved Protein Motifs Prediction

In total, ten conserved protein motifs were explored in GUS protein of nineteen bacteria through MEME. The number of amino acids were found to be 29 (motifs 1, 5, 6, 8, 9 and 10), 50 (motif 2), 41 (motif 3 and 4) and 25 (motif 7). The locations of these motifs with their *p*-values are shown (Figure 6). The motif results were found to be significant with *p*-value < 0.05 in the case of all bacteria except *S. caeli*. Sequences, E-values, site count, width, relative entropy and bayes threshold are also mentioned (Figure 7). E-value is an estimate of the expected number of motifs with the given log likelihood ratio (or higher) and with the same width and site count, that one would find in a similarly sized set of random sequences. The E-values for all the ten motifs were found to be significant i.e., <0.05. Site count is the number of sites contributing to the construction of motif. The maximum number of site count, i.e., 19, was observed in the case of third, fifth and sixth motif sequences. The lowest number of site count, i.e., 13, was observed in case of second and ninth motifs. The width of the motif describes a pattern of a fixed width as no gaps are allowed in MEME motifs. The width was observed in the range of 25–50. Maximum was predicted in second motif and minimum in the case of first, fifth, sixth, eighth, ninth and tenth motifs. The conserved proteins motifs explored via MEME were subjected to Interproscan to predict their molecular and biological functions. This revealed association of these motifs with carbohydrate metabolism and hydrolyses of O-glycosyl compounds.

### 3.7. Antigenic Peptide Prediction

Positions and number of sequences that might be involved in antigenic propensity were different in all proteins i.e., Lacticaseibacillus rhamnosus (24), Roseburia intestinalis (27), Coprococcus comes (30), Faecalibacterium prausnitzii 1 (25), Faecalibacterium prausnitzii 2 (29), Faecalibacterium prausnitzii 3 (27), Propionibacterium acnes (22), Methylococcaceae bacterium (23), Staphylococcus xylosus (19), Streptococcus suis (20), Bacillus sp. M4U3P1 (22), Escherichia coli (strain K12) (26), Sphingomonas aquatilis NBRC 16722 (27), Citrobacter amalonaticus (28), Enterococcus gallinarum 1 (26), Enterococcus gallinarum 2 (27), Salmonella enterica (27), Staphylococcus caeli (22), Staphylococcus haemolyticus (23). The antigenicity plots of GUS proteins are indicated in Figure 8.

## 4. Discussion

The first verification of microbiota presence and dysbiosis in breast cancer tissue has been reported through next generation sequencing (NGS) analysis of breast tumor tissue as well as the normal tissue adjacent to tumor. Qualitative analysis revealed the presence of *Methylobacterium radiotolerans* and *Sphingomonas yanoikuyae* in tumor and normal tissues, respectively. Quantitative PCR-based analysis showed reduced load of bacterial DNA in cancerous tissue proving the breast cancer association with dysbiosis [11]. Multiple experimental evidences of the role of microbial GUS enzyme in breast cancer has been reported in the literature. In a study, the potential of 35 GUS enzymes to reactivate glucuronidated estrogen was explored using in-vivo, in-vitro and in-fimo techniques. It was found that GUS enzymes belonging to classes L1, ML1 and FMN were very active in the activation of conjugated estrogens [28]. The association of gmGUS with estrobolome has also been studied via the inspection of estrogen replacement therapy impact on GUS enzymes of GIT microbiota and the microbial composition. Long-term exposure to ERT altered the microbial composition of GIT accompanied with reduced GUS enzyme activities. ERT induced dysbiosis by reducing the number of *L. rhamnosus*, *F. prausnitzii* and enhancing the *R. gnavus* [25,26,27,28,29,30,31,32,33,34,35,36,37,38,39,40,41,42,43,44,45,46,47,48]. 

The gmGUS play their role in estrobolome by reversing the glucuronidation process and catalyze estrogens activation by breaking glucuronic moiety. Estrogen and estrone glucuronides that might be the substrates for gmGUS include 17-α estradiol 17-O, 17-β estradiol 17-O, 17-β estradiol 3-O, 2-hydroxy-17-β estradiol 2-O, 2-hydroxy-17-β estradiol 3-O, 4-hydroxy 17-β estradiol-3-O, 4-hydroxy 17-β estradiol-4-O, estrone 3-O, 2-hydroxy estrone-2-O, 2-hydroxy estrone-3-O, 4-hydroxy estrone-3-O, 16-α-17-β estriol 3-O, 16-β-17-β estriol 3-O, 16-α-17-α estriol 3-O, 16-α-17-β estriol 16-O, 16-α-17-α estriol 16-O, 16-β-17-β estriol 16-O, 16-α-17-β estriol-16-O, 16-α-17-α estriol-17-O and 16-β-17-β estriol-17-O, respectively [30].

This reaction releases aglycones. The process of deconjugation occurs as estrogens after GIT via bile [24]. Estrogens without deconjugation, due to high polarity and hydrophilicity, are dissolved in blood and are removed through the kidneys as urine. However, on deconjugation, estrogens via reabsorption in mucosa enter the portal vein [32]. Due to the association of high estrogen concentration with breast cancer gut microbes and breast cancer axis has been as emerging research area. 

With reference to sub-cellular localization and physicochemical properties, bacteria in breast cancer patients were found to be more diverse as compared to those reported in normal tissues. The diversity of GUS enzyme found in the present study is consistent with the literature [49]. This protein in *Ruminococcus gnavus* has been reported to show similarity of 69%, 61%, 59% and 58% with *L. gasseri*, *E. coli*, *C. perfringens* and *S. aureus*, respectively [50]. Therefore, this study has further strengthened the diversified nature of GUS enzyme.

The GUS protein in *S. xylosus*, *S. suis*, *S. aquatilis* NBRC 16722, *C. amalonaticus*, *S. caeli* and *E. gallinarum* 1 exhibited extreme values of pI, extinction coefficient, instability index and aliphatic index. Among the probiotics of normal tissue, only *L. rhamnosus* GUS protein showed extreme value for aliphatic index. The total number of amino acids was found to be in the range of 596–795 with variable molecular weights. The isoelectric point analysis revealed that GUS of *S. aquatilis* NBRC 16722 was alkaline with pI value 8.92 while in all other cases, GUS showed acidic pI. The pI value for *S. caeli* is consistent with earlier reported pI value for *E. coli* HGU-3 [51]. Literature reports that GUS enzyme activity increases at high pH and causes cancer [52]. According to this information, in the present study GUS enzyme of *S. aquatilis* showed alkaline nature as per its pI. While enzymes of all other bacteria except *S. caeli* exhibited low acidic pI, which might have some association with the cancer-causing potential of this enzyme.

Aliphatic index reflects the relative volume of protein occupied by aliphatic side chain containing amino acids which indicates increased thermostability [53]. As all the bacterial proteins in the present study showed the range of 71.01–88.10 so GUS protein is found to be highly thermostable. Instability index is a measure of protein stability in test tube [54]. The value below 40 indicates stability of protein, so GUS enzyme in the case of all present study bacteria except C. amalonaticus is considered to be highly stable. 

The 3-D configuration was generated using PHYRE2 tool and further verified by plotting Ramachandran plot using SAVES server. Except in the *R. intestinalis* and *F. prausnitzii* 1, GUS enzyme in all other bacteria showed marked variation in 3D structures. GUS enzymes from nineteen microbiota showed high variation with regard to physicochemical properties, sub-cellular localization, 3D configuration and antigenic sites so this is a highly diverse protein.

The catalytic pockets were identified along with different parameters of tunnels. Druggability is the potential of a molecule of being controlled by therapeutic drugs [55]. It is the measurement of binding affinity of catalytic site to drug like organic molecule of host [56]. The druggability values for catalytic sites of GUS enzyme are reflecting their challenging nature and do not prove them as excellent drug targets. Bottle neck radius is the measure of maximum size of probe that can fit in the narrowest portion of tunnel. Length of tunnel measures the distance between starting point and protein surface. In the present study, tunnel length was measured in the range of 1.4 to 15.9. Curvature shows the shape of tunnel which is the ratio of tunnel length and shortest distance between the starting and ending points of tunnel. In GUS enzymes, tunnel curvature was measured in the range of 0.65 to 1.4. Length and curvature also gives an idea about substrate specificity of the enzyme. Tunnels geometry may affect catalysis via channeling of substrate [57]. However, these catalytic sites can be engineered for targeting by drugs to inhibit the estrogen reactivation in breast cancer patients. The attributes of tunnel can be useful in drug designing experiments in future.

Conserved protein motifs are those sequences of proteins that undergo small variations with time. The variations might involve substitutions of fewer amino acids and replacement with amino acid having similar biochemical properties. These motifs play crucial roles in the stability and formation of catalytic site of protein. Conserved protein motifs identified in present study bacteria also show their relatedness [58].

Antigenic peptides are the bacterial proteins which directly interact with host immune system; therefore, they can be good candidates for the development of vaccines [59]. GUS enzyme in all the nineteen bacteria showed antigenic peptides in the range of 22–29. The presence of these large numbers of antigenic peptides gives a clue about the possible immunomodulatory role of these bacteria. Due to the variety in antigenic peptides, catalytic sites and the adjacent loop structures, anti-cancer medication therapy can be managed via GUS enzyme inhibition also reported in literature [49]. High concentration and deglucuronidation potential of GUS enzyme in breast cancer patients can also be used for bioactivation of glucuronide anticancer prodrugs [60].

## 5. Conclusions

Identification of the structural properties and present study findings regarding estrogen-reactivating protein (GUS enzyme) in bacteria found in normal and breast cancer patients might provide us multiple directions for modification of this enzyme. As it is very easy to perform manipulations successfully at the level of probiotics as compared to human genes, the manipulation at probiotics level might be helpful in reducing breast cancer risk through inhibiting reactivation of this estrobolome-associated protein. Additionally, the active sites explored in the present study can be inspected further for their possible role in deglucuronidation of glucuronidated estrogens. Exploration of these catalytic sites might help in modification of GUS enzyme to prevent its estrogen deconjugation potential. Modifications may include alteration in the structure of active sites participating in deglucuronidation by inducing mutations at specific points in uidA gene and deletion of conserved protein motifs. For these modifications, site-directed mutagenesis can be performed which may lead to destabilization of protein, thus inhibiting its estrogen reactivation potential.

## Figures and Tables

**Figure 1 genes-13-01545-f001:**
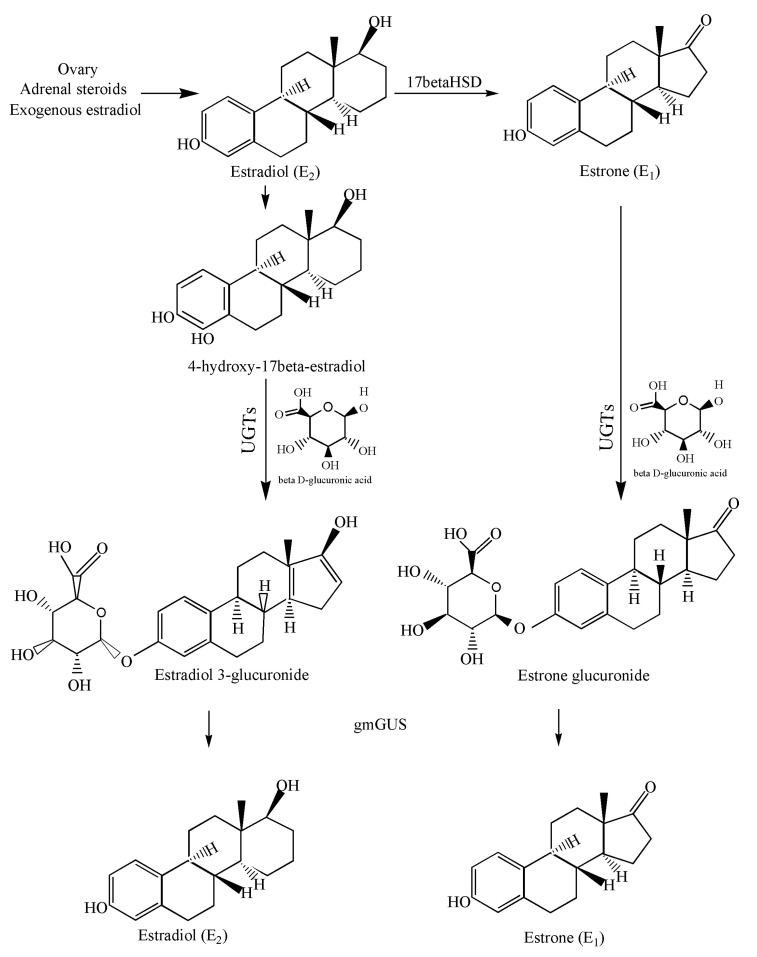
Pathway involved in conjugation and deconjugation of estradiol and estrone by UGTs and gmGUS, respectively.

**Figure 2 genes-13-01545-f002:**
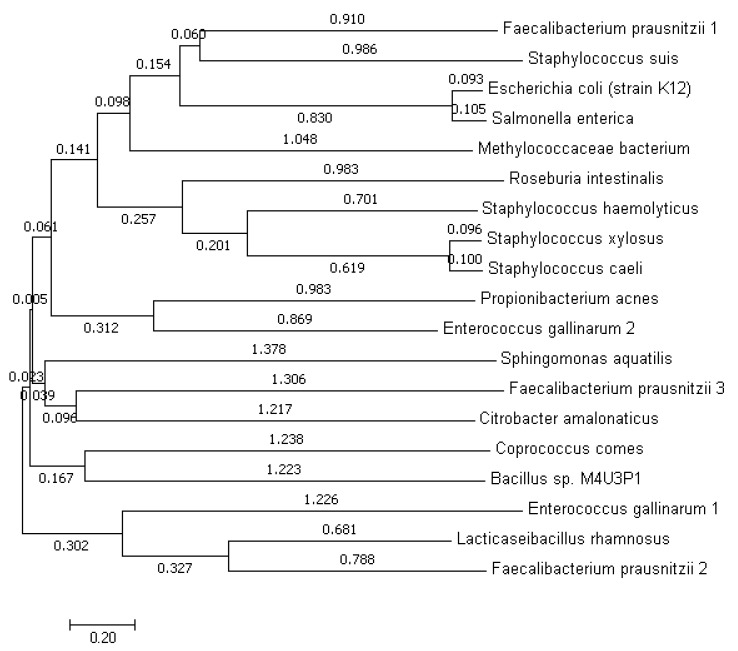
Phylogenetic tree based on GUS protein sequences of present study bacteria constructed using neighbor-joining method The optimal tree with the sum of branch length = 19.62208961 is shown. The tree is drawn to scale, with branch lengths (next to the branches) in the same units as those of the evolutionary distances used to infer the phylogenetic tree.

**Figure 3 genes-13-01545-f003:**
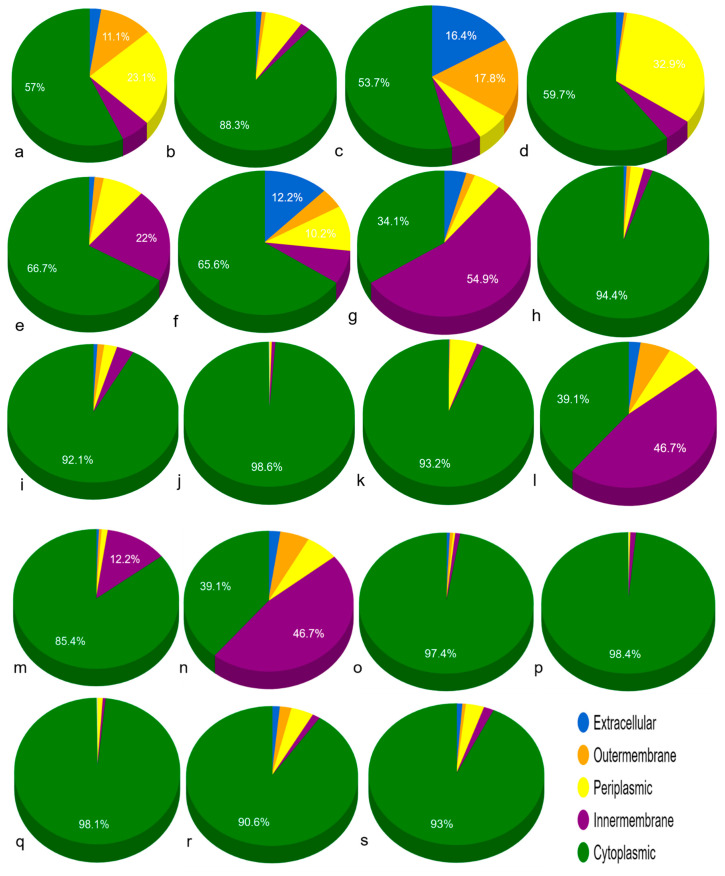
Sub-cellular localization of GUS proteins in bacteria found in normal human and breast cancer patients GIT (**a**) *L. rhamnosus*, (**b**) *R. intestinalis*, (**c**) *C. comes*, (**d**) *F. prausnitzii* 1, (**e**) *F. prausnitzii* 2, (**f**) *F. prausnitzii* 3, (**g**) *P. acnes*, (**h**) *M. bacterium*, (**i**) *S. xylosus*, (**j**) *S. suis*, (**k**) *Bacillus* sp. M4U3P1, (**l**) *E. coli* (strain K12), (**m**) *S. aquatilis* NBRC 16722, (**n**) *C. amalonaticus*, (**o**) *E. gallinarum* 1, (**p**) *E. gallinarum* 2, (**q**) *S. enterica*, (**r**) *S. caeli*, (**s**) *S. haemolyticus*.

**Figure 4 genes-13-01545-f004:**
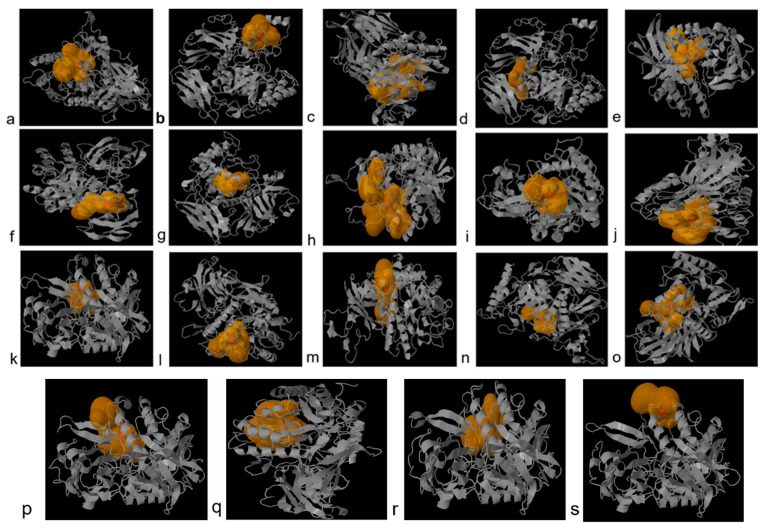
Catalytic sites predicted using Caver Web server in present study bacteria (**a**) *L. rhamnosus*, (**b**) *R. intestinalis*, (**c**) *C. comes*, (**d**) *F. prausnitzii* 1, (**e**) *F. prausnitzii* 2, (**f**) *F. prausnitzii* 3, (**g**) *P. acnes*, (**h**) *M. bacterium*, (**i**) *S. xylosus*, (**j**) *S. suis*, (**k**) *Bacillus* sp. M4U3P1, (**l**) *E. coli* (strain K12), (**m**) *S. aquatilis* NBRC 16722, (**n**) *C. amalonaticus*, (**o**) *E. gallinarum* 1, (**p**) *E. gallinarum* 2, (**q**) *S. enterica*, (**r**) *S. caeli*, (**s**) *S. haemolyticus*.

**Figure 5 genes-13-01545-f005:**
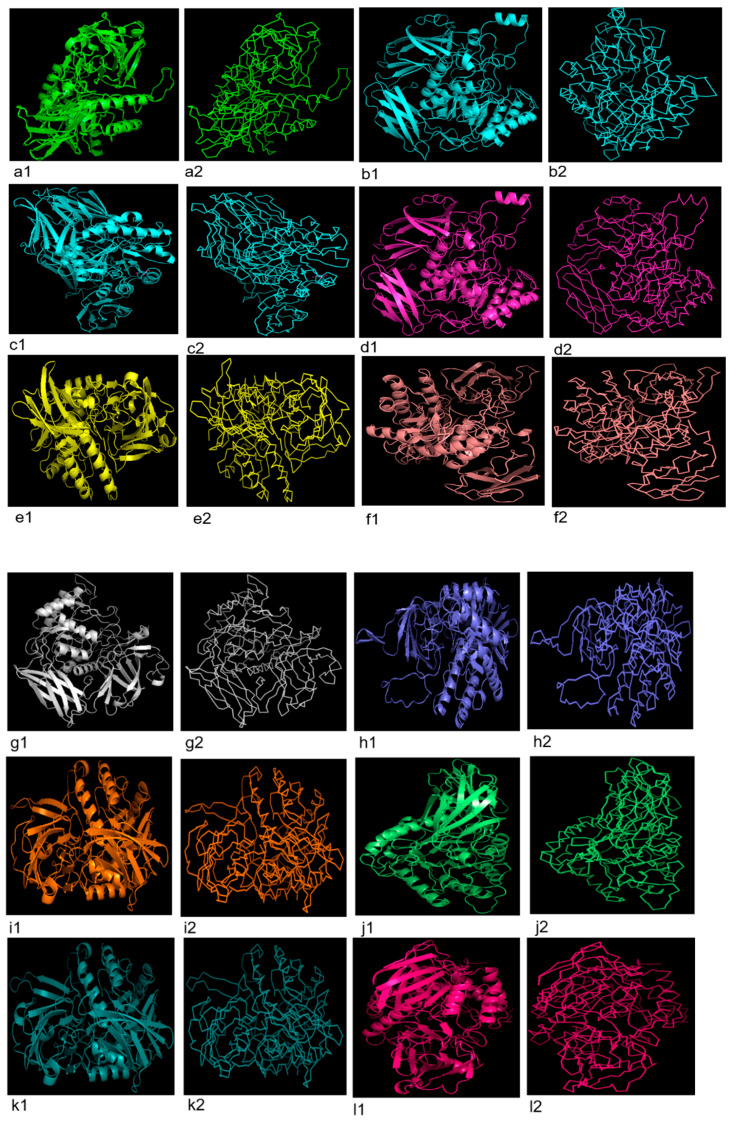
Three dimensional configuration (cartoon and ribbon form) of uidA protein predicted using PHYRE2 tool and visualized by PyMOL, in probiotics documented in present study**.** (**a1**,**a2**) *L. rhamnosus*, (**b1**,**b2**) *R. intestinalis*, (**c1**,**c2**) *C. comes*, (**d1**,**d2**) *F. prausnitzii* 1, (**e1**, **e2**) *F. prausnitzii* 2, (**f1**,**f2**) *F. prausnitzii* 3, (**g1**,**g2**) *P. acnes*, (**h1**,**h2**) *M. bacterium*, (**i1**,**i2**) *S. xylosus*, (**j1**,**j2**) *S. suis*, (**k1**,**k2**) *Bacillus* sp. M4U3P1, (**l1**,**l2**) *E. coli* (strain K12), (**m1**,**m2**) *S. aquatilis* NBRC 16722, (**n1**,**n2**) *C. amalonaticus*, (**o1**,**o2**) *E. gallinarum* 1, (**p1**,**p2**) *E. gallinarum* 2, (**q1**,**q2**) *S. enterica*, (**r1**,**r2**) *S. caeli*, (**s1**,**s2**) *S. haemolyticus*.

**Figure 6 genes-13-01545-f006:**
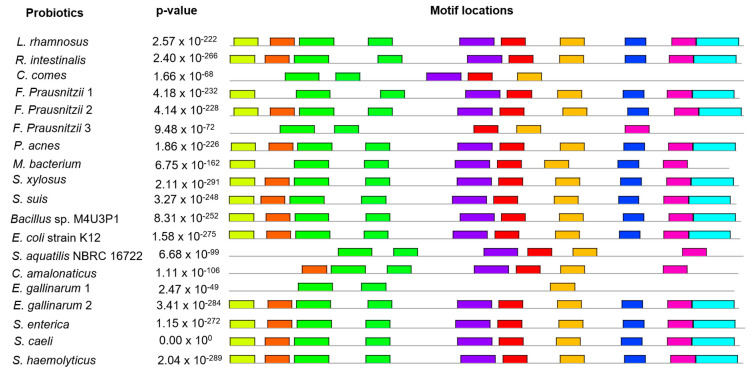
Location of motif sites with their corresponding combined match *p*-value for uidA protein of bacteria, predicted by MEME suite. Each block represents the position and strength of a motif site. The height of a block gives an indication of the significance of the site as taller blocks are more significant. The height is calculated to be proportional to the negative logarithm of the *p*-value of the site, truncated at the height for *p*-value of 1 × 10^−10^. Combined match *p*-value is defined as the probability that a random sequence (with the same length and conforming to the background) would have position *p*-values such that the product is smaller or equal to the value calculated for the sequence under test. Ten different colors are used to depict the ten different motifs.

**Figure 7 genes-13-01545-f007:**
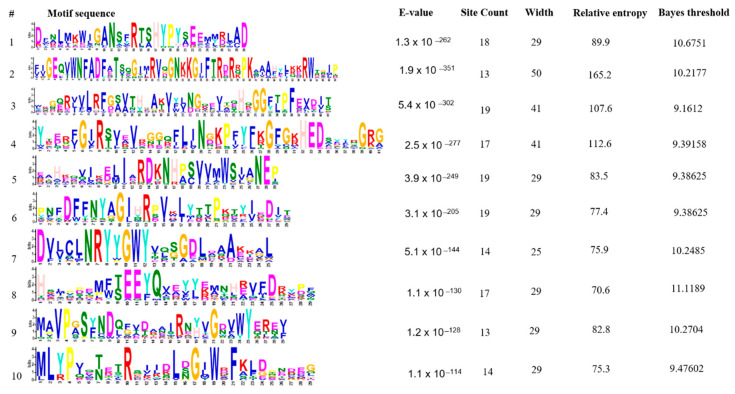
Sequences, E-values, site count, width, relative entropy and bayes threshold of conserved motifs of uidA protein predicted in probiotics documented in present study E-value show the statistical significance of the motif. It is an estimate of the expected number of motifs with the given log likelihood ratio (or higher) and with the same width and site count that one would find in a similarly sized set of random sequences. Site count is the number of sites contributing to the construction of the motif. The width of the motif describes a pattern of a fixed width as no gaps are allowed in MEME motifs.

**Figure 8 genes-13-01545-f008:**
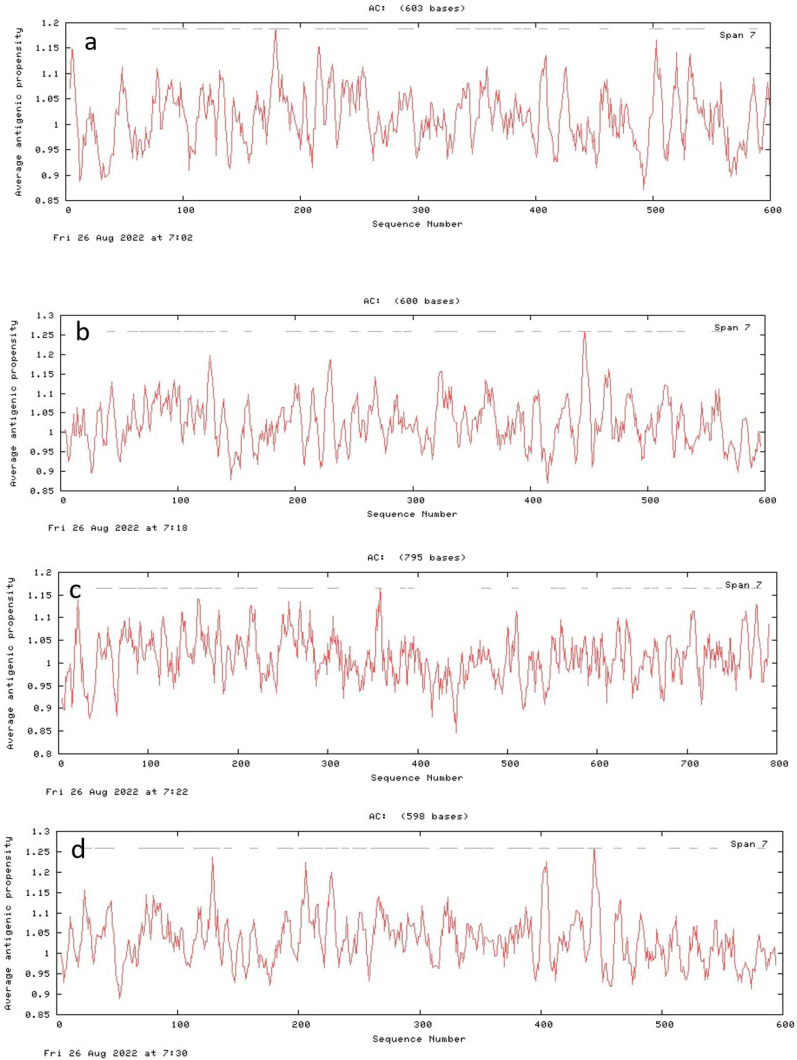
Antigenecity plots for uidA proteins of probiotics (**a**) *L. rhamnosus*, (**b**) *R. intestinalis*, (**c**) *C. comes*, (**d**) *F. prausnitzii* 1, (**e**) *F. prausnitzii* 2, (**f**) *F. prausnitzii* 3, (**g**) *P. acnes*, (**h**) *M. bacterium*, (**i**) *S. xylosus*, (**j**) *S. suis*, (**k**) *Bacillus* sp. M4U3P1, (**l**) *E. coli* (strain K12), (**m**) *S. aquatilis* NBRC 16722, (**n**) *C. amalonaticus*, (**o**) *E. gallinarum* 1, (**p**) *E. gallinarum* 2, (**q**) *S. enterica*, (**r**) *S. caeli*, (**s**) *S. haemolyticus*.

**Table 1 genes-13-01545-t001:** Physicochemical properties of GUS enzymes in bacteria found in normal and breast cancer patients.

#	Probiotics	No. of Amino Acids	Mol. Wt.	pI	Half-Life (Hours)	Extinction Coefficient	Instability Index	Aliphatic Index
1	*L. rhamnosus*	603	68,570.92	5.35	30	101,760	30.86	71.01
2	*R. intestinalis*	600	68,337.66	5.06	30	109,140	33.60	73.75
3	*C. comes*	795	87,709.17	5.85	30	126,920	26.87	78.06
4	*F. prausnitzii* 1	598	67,535.00	5.34	30	119,345	35.75	74.03
5	*F. prausnitzii* 2	608	69,345.36	6.13	30	118,970	33.77	75.41
6	*F. prausnitzii* 3	639	72,500.39	5.92	30	131,960	32.82	71.85
7	*P. acnes*	594	67,236.34	5.12	30	125,040	33.75	78.94
8	*M. bacterium*	586	66,604.06	5.91	30	131,125	36.17	81.98
9	*S. xylosus*	597	67,983.63	5.21	30	89,980	31.77	82.84
10	*S. suis*	599	68,912.07	5.11	30	95,355	29.20	88.10
11	*Bacillus* sp. M4U3P1	604	68,912.10	5.17	30	100,855	37.19	80.98
12	*E. coli* (strain K12)	603	68,447.00	5.24	30	140,760	26.68	77.74
13	*S. aquatilis* NBRC 16722	631	69,237.47	8.92	30	124,455	37.69	83.88
14	*C.* *amalonaticus*	601	69,674.51	5.33	30	119,805	40.93	80.80
15	*E. gallinarum* 1	596	69,632.94	5.01	30	155,075	25.88	75.87
16	*E. gallinarum* 2	602	69,494.74	5.05	30	106,480	30.79	85.12
17	*S. enterica*	603	68,922.61	5.45	30	136,750	30.39	77.78
18	*S. caeli*	597	67,893.91	4.91	30	92,960	30.87	81.76
19	*S. haemolyticus*	601	68,750.54	5.46	30	92,835	32.41	81.20

**Table 2 genes-13-01545-t002:** Sub-cellular localization of GUS enzymes in the present study bacteria predicted on the basis of CELLU2GO tool.

Bacteria	Sub-Cellular Localization				
	Extracellular	Outer Membrane	Periplasmic Space	Inner Membrane	Cytoplasm
Bacteria associated with normal individuals					
*L. rhamnosus*	0.176	0.779	1.619	0.433	3.993
*R. intestinalis*	0.091	0.057	0.525	0.148	6.180
*C. comes*	1.146	1.249	0.448	0.396	3.761
*F. prausnitzii* 1	0.111	0.038	2.304	0.365	4.183
*F. prausnitzii* 2	0.084	0.126	0.579	1.541	4.669
*F. prausnitzii* 3	0.857	0.311	0.713	0.530	4.589
*P. acnes*	0.030	0.010	0.232	0.230	6.498
Bacteria associated with breast cancer patients					
*M. bacterium*	0.298	0.112	0.360	3.839	2.390
*S. xylosus*	0.050	0.054	0.174	0.117	6.604
*S. suis*	0.063	0.089	0.169	0.229	6.449
*Bacillus* sp. M4U3P1	0.004	0.008	0.038	0.049	6.901
*E. coli* (strain K12)	0.019	0.012	0.353	0.090	6.526
*S. aquatilis* NBRC 16722	0.161	0.389	0.440	3.271	2.738
*C.* *amalonaticus*	0.043	0.036	0.083	0.857	5.982
*E. gallinarum* 1	0.052	0.041	0.026	0.066	6.815
*E. gallinarum* 2	0.011	0.008	0.021	0.074	6.885
*S. enterica*	0.008	0.007	0.081	0.038	6.865
*S. caeli*	0.108	0.152	0.296	0.102	6.343
*S. haemolyticus*	0.082	0.044	0.236	0.131	6.508

**Table 3 genes-13-01545-t003:** Secondary structure for GUS proteins of bacteria predicted using SOPMA tool.

Probiotics	α Helix (%)	Extended Strand (%)	β Turn (%)	Random Coil (%)
Normal tissue associated bacteria				
*L. rhamnosus*	26.37	23.55	7.13	42.95
*R. intestinalis*	27.33	22.67	6.50	43.50
*C. comes*	23.02	25.91	8.30	42.77
*F. prausnitzii* 1	27.42	22.74	6.69	43.14
*F. prausnitzii* 2	26.15	22.04	6.25	45.56
*F. prausnitzii* 3	23.32	25.82	8.29	42.57
*P. acnes*	26.60	24.75	6.90	41.75
Breast cancer patients associated bacteria				
*M. bacterium*	26.45	23.21	6.31	44.03
*S. xylosus*	26.80	23.95	6.87	42.38
*S. suis*	26.88	22.37	6.84	43.91
*Bacillus* sp. M4U3P1	27.32	22.52	6.62	43.54
*E. coli* (strain K12)	25.70	23.88	7.13	43.28
*S. aquatilis* NBRC 16722	22.66	24.25	5.39	47.70
*C. amalonaticus*	26.79	24.13	5.49	43.59
*E. gallinarum* 1	25.17	23.49	6.21	45.13
*E. gallinarum* 2	27.24	23.59	7.14	42.03
*S. enterica*	26.37	24.21	6.63	42.79
*S. caeli*	27.81	23.62	6.70	41.88
*S. haemolyticus*	26.46	24.13	6.99	42.43

**Table 4 genes-13-01545-t004:** Relevance score, druggability, bottleneck radius, length and curvature of catalytic sites tunnels determined in present study bacteria using Caver Web tool.

Bacteria	Relevance Score (%)	Druggability	Bottleneck Radius (Angstrom)	Length (Angstrom)	Curvature
Normal tissue associated microbiota					
*L. rhamnosus*	100	0.14	2.3	1.4	1.1
*R. intestinalis*	100	0.17	1.6	4.3	1.2
*C. comes*	100	0.28	1.9	2.7	1.3
*F. prausnitzii* 1	100	0.07	2.1	2.5	1.0
*F. prausnitzii* 2	100	0.07	1.6	8.8	1.1
*F. prausnitzii* 3	100	0.05	1.2	10.7	1.4
*P. acnes*	100	0.45	1.2	10.7	1.4
Breast cancer patients associated bacteria					
*Methylococcaceae* bacterium	100	0.80	2.3	6.9	1.1
*S. xylosus*	100	0.60	2.0	5.7	1.1
*S. suis*	100	0.07	2.1	1.5	1.0
*Bacillus* sp. M4U3P1	100	0.07	2.2	4.4	1.2
*E. coli* (strain K12)	100	0.06	2.2	1.9	1.2
*S. aquatilis* NBRC 16722	100	0.06	1.1	8.7	0.65
*C. amalonaticus*	100	0.14	1.3	14.6	1.2
*E. gallinarum* 1	100	0.13	1.2	15.9	1.3
*E. gallinarum* 2	100	0.23	2.8	2.0	1.2
*S. enterica*	100	0.51	2.5	1.5	1.0
*S. caeli*	100	0.39	2.3	4.0	1.1
*S. haemolyticus*	100	0.52	1.9	3.4	1.0

**Table 5 genes-13-01545-t005:** Interpretation of Ramachandran plots for GUS proteins of bacteria.

Bacteria	Residues in Most Favored Region (%)	Residues in Additional Allowed Regions (%)	Residues in Generously Allowed Regions (%)	Residues in Disallowed Regions (%)	G-Factors		
					Dihedrals	Covalent	Overall
Normal tissue associated bacteria							
*L. rhamnosus*	88.4	10.6	0.8	0.2	−0.13	0.47	0.11
*R. intestinalis*	85.2	14	0.2	0.6	0.07	0.42	0.22
*C. comes*	88.2	10.0	1.1	0.6	−0.11	0.28	0.05
*F. prausnitzii* 1	86.3	13.3	0.2	0.2	0.07	0.48	0.24
*F. prausnitzii* 2	91.3	7.9	0.6	0.2	−0.10	0.49	0.14
*F. prausnitzii* 3	81.7	16.7	0.9	0.7	−0.26	0.37	0.00
*P. acnes*	82.1	16.9	0.4	0.6	−0.23	0.42	0.03
Breast cancer patients associated bacteria							
*Methylococcaceae* bacterium	88.2	10	1.0	0.8	−0.14	0.37	0.07
*S. xylosus*	86.8	11.5	0.9	0.8	−0.05	0.57	0.20
*S. suis*	87.5	11.7	0.6	0.2	−0.00	0.62	0.24
*Bacillus* sp. M4U3P1	87.4	11.7	0.6	0.4	−0.04	0.57	0.20
*E. coli* (strain K12)	87.3	11.3	0.9	0.4	−0.11	0.24	0.04
*S. aquatilis* NBRC 16722	83.6	14.6	1.4	0.4	−0.32	−0.06	−0.21
*C. amalonaticus*	86.4	12.0	1.1	0.4	−0.14	0.47	0.10
*E. gallinarum* 1	84.9	10.2	3.1	1.8	−0.08	0.31	0.08
*E. gallinarum* 2	86.5	11.3	1.3	0.9	−0.02	0.57	0.21
*S. enterica*	86.6	11.3	1.3	0.9	−0.02	0.57	0.21
*S. caeli*	86.6	11.7	1.1	0.6	−0.12	0.25	0.04
*S. haemolyticus*	85.9	12.1	0.9	1.1	−0.05	0.55	0.19

## Data Availability

The sequences of bacterial proteins documented in present study are available at Uniprot (https://www.uniprot.org) database.

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
