# Peer review of "In-Silico Characterization of Estrogen Reactivating β-Glucuronidase Enzyme in GIT Associated Microbiota of Normal Human and Breast Cancer Patients"

_genes, 2022, doi:10.3390/genes13091545_

Round 1

Reviewer 1 Report

This article study to explore GIT microbiome encoded GUS enzymes (GUSOME) repertoire in normal human and breast cancer patients. This study will help to facilitate helpful in modifications of GIT microbiota for reducing reactivation of estrogen thus preventing the breast cancer incidence. Before recommending this article for publication, there are some shortcomings for that should be resolve.

General comments

Overall, the study is well designed and presented in a good way, but mostly the literature is not cited and contain grammatical mistakes.  

Abstract

Line 19-20 full form of the abbreviations must be added.

The methods and results must be specified and briefly explained.

 Also add quantitative results in this section.

Introduction

The introduction part is well written but still some details are required.

Provide economic and commercial importance of the Probiotic.

Line 41 to 45 must be cited with relevant literature.

 https://doi.org/10.3390/microorganisms10050954,

Add mechanism of GUS enzymes (GUSOME) and its role in breast cancer.

Line 94-95 must be cited.

Materials and methods

Section 2.1 must be cited with relevant study. https://doi.org/10.3390/ijms22179175,  

Provide complete details of section 2 phylogenetic analysis.

Section 2.6. and 2.5. If the authors provided 3d structure, then why 2d structure is required. Add details.

Details of the section 2.7 is missing should be added

Add purpose of section 2.8 and other sections.

Results

Section 3.1 looks like methodology. The authors are directed to discuss phylogenetic tree.

On which basis these strains were selected for phylogenetic analysis.

Section 3.3 add figure or table.

Writing on the yellow color is not visible change the color.

Section 3.7 generic name must be abbreviated, Like L. rhamnosus.

Provide details of figure 7 in section 3.7.

Mostly results are poorly discussed. These can be consulted with the professional such as phylogenetic and motif results.

Discussion

The authors should provide relation of the GUS enzymes (GUSOME) genes and its associated pathways with the help of related previous studies. Also compare the results of these studies with the current study.

Mainly the discussion is not cited.

The authors are directed to cite the relevant literature.

Conclusion

In conclusion also recommend techniques which can be helpful for the manipulation of these enzymes and its related genes.  

Author Response

Rebutal letter

Comment 1 (General comments)

Overall, the study is well designed and presented in a good way, but mostly the literature is not cited and contain grammatical mistakes.  

Reply to Reviewer comment

The literature and grammatical errors have been improved (Please see the reply to reviewers comments below).

Comment 2

Line 19-20 full form of the abbreviations must be added.

Reply to Reviewer comment

The abbreviations full form has been incorporated.

Comment 3

The methods and results must be specified and briefly explained.  Also add quantitative results in this section.

Reply to Reviewer comment

The abstract has been modified as follows;

Protein sequences of enzymes retrieved from UniProt database were subjected to ProtParam, CELLO2GO, SOPMA (secondary structure prediction method), PDBsum (Protein Database summaries), PHYRE2 (Protein Homology/AnalogY Recognition Engine), SAVES v6.0 (Structure Validation Server), MEME version 5.4.1 (Multiple Em for Motif Elicitation), Caver Web server v 1.1, Interproscan and Predicted Antigenic Peptides tool. Analysis revealed the number of amino acids, isoelectric point, extinction coefficient, instability index and aliphatic index of GUS enzymes in the range of 586-795, 4.91-8.92, 89980-155075, 25.88-40.93 and 71.01-88.10, respectively. Sub-cellular localization of enzyme was restricted to cytoplasm and inner-membrane in case of breast cancer patients bacteria as compared to periplasmic space, outer membrane and extracellular space in normal GIT bacteria. The 2-D structure analysis showed alpha helix, extended strand, beta turn and random coil in the range of 27.42 - 22.66%, 22.04 - 25.91 %, 5.39 – 8.30 % and 41.75 – 47.70%, respectively. Druggability score was found 0.05 – 0.45 and 0.06 – 0.80 in normal and breast cancer patients GIT, respectively. The radius, length and curvature of catalytic sites were observed to be 1.1 – 2.8 Å, 1.4 – 15.9 Å and 0.65 – 1.4, respectively. Ten conserved protein motifs with p < 0.05 and width 25 – 50 were found. Antigenic propensity associated sequences were 20 – 29. Present study findings hints about use of the bacterial GUS enzymes against breast cancer tumor after modifications via site-directed mutagenesis of catalytic sites

Comment 4

Provide economic and commercial importance of the Probiotic.

Reply to Reviewer comment

The economic and commercial importance of the probiotics has been incorporated in introduction in second paragraph.

Due to health benefits, probiotics contribute to the development of commercially important functional foods and pharmaceutical formulations. Like L. rhamnosus, L. acidophilus, L. casei, L. plantarum and Bifidobacterium species are commonly used in dairy food production including fermented milk, yogurt, cheese and baby milk powder. Probiotics including L. Rosell 52, Bifidobacterium, S. cerevisiae, L. salivarius, L. reuteri and L. acidophilus NCFB 1748 have potential applications in pharmacy due to their association with prevention of pathogens, travelers diarrhea and antibiotic associated diarrhea, reduction of upper respiratory tract infections and constipation improvement etc.

Comment 5

Line 41 to 45 must be cited with relevant literature.  https://doi.org/10.3390/microorganisms10050954,

Reply to reviewers comment

Bacillus, Enterobacteriaceae, Staphylococcus and Escherichia coli have been reported in high abundance in breast cancer patients. Histone-2AX (H2AX) phosphorylation assay performed on HeLa cells showed role of these bacteria in DNA double stranded breaks [1]. A study initiated to analyze breast cancer associated dysbiosis in pre-menopausal women using targeted metabolomics, 16S rRNA sequencing and cell culture methods confirmed the presence of Fusobacterium nucleatum, Pediococcus, Salmonella enterica, Corynbacterium, Shewanella putrefaciens, Enterococcus gallinarum and Thermus scotoductus and Desulfovibrio [2]. Ralstonia genus has been found in breast cancer tissue through analyses of variable regions of 16S rRNA [3].

Comment 6

Add mechanism of GUS enzymes (GUSOME) and its role in breast cancer.

Reply to reviewers comment

Role and mechanism is already mentioned in paragraph 5.

Comment 7

Line 94-95 must be cited.

Reply to reviewers comment

The line numbers 94 and 95 have been cited.

2D structure  

            For secondary structure prediction, SOPMA tool from Network Protein Sequence Analysis (https://npsa-prabi.ibcp.fr/cgi-bin/npsa_automat.pI?page=npsa_sopma.html, accessed on 30 July 2022,) was employed [4]. To predict secondary motif map PDBsum tool (http://www.ebi.ac.uk/thornton-srv/databases/cgibin/pdbsum/GetPage.pl?pdbcode=index.html, accessed on 30 July 2022) was used. To predict the catalytic site of GUS enzyme Caver Web server v 1.1 (https://loschmidt.chemi.muni.cz/caverweb/, accessed on 4 August 2022) was used [5].

Comment 8

Section 2.1 must be cited with relevant study. https://doi.org/10.3390/ijms22179175,  

Reply to reviewers comment

To retrieve protein sequences of bacteria documented in present project, Uniprot database (https://www.uniprot.org, accessed on 21-24 July 2022) was explored [6]. Sequences retrieved, their accession numbers and bacterial species selected for this study are mentioned (Supplementary Data Table 1).

Comment 9

Provide complete details of section 2 phylogenetic analysis.

Reply to reviewers comment

            To construct the phylogenetic tree initially protein sequences of nineteen bacteria documented in present study were aligned using Clustal Omega Multiple Sequence Alignment Tool [7]. The aligned file was then subjected to MEGA version 7 [8]. The evolutionary history was inferred using the Neighbor-Joining method [9]. The evolutionary distances were computed using the Poisson correction method and are in the units of the number of amino acid substitutions per site [10]. All positions containing gaps and missing data were eliminated. There were a total of 586 positions in the final dataset.

Comment 10

Section 2.6. and 2.5. If the authors provided 3d structure, then why 2d structure is required. Add details.

Reply to reviewers comment

The 25 to 75 % of protein is made up of secondary structure building blocks [11]. Alpha helix, extended strand, beta turn and bends are the basic elements of secondary conformation. It is important to analyze the impact of SNPs on these elements in order to get an insight into the deleteriousness of SNPs. While 3D configuration has been analyzed to study the effect of SNPs on overall folding and shape of protein.

This information is incorporated in sub-section 2.5.

Comment 11

Details of the section 2.7 is missing should be added

Reply to reviewers comment

Following information has been added in section 2.7.

To predict the conserved motifs in protein sequences of probiotics, MEME version 5.4.1 (http://meme.sdsc.edu/meme/meme.html, accessed on 26 July 2022) was used. This tool usually finds three motifs by default however, in present study we tried to find upto ten motifs. All other parameters were set according to default settings. To estimate the ontology of each individual conserved domain, Interproscan (http://www.ebi.ac.uk/interpro/search/sequence/, accessed on 4 August 2022) was employed.

Comment 12

Add purpose of section 2.8 and other sections.

Reply to reviewers comment

The purpose of each section including is already included in the start of each section and is highlighted yellow. For reviewer’s kind information, the purpose has been given below.

Section

Purpose

Section 2.1

(UniProt database)

Retrieval of beta-glucuronidase sequences alongwith the accession numbers, belonging to present study bacteria.

Section 2.2

To perform multiple alignment of beta glucuronidase enzymes in all the present study bacteria

Clustal Omega Multiple Sequence Alignment Tool

MEGA version 7

Multiple sequence alignment is used to construct phylogenetic tree

Section 2.3

(ProtParam)

Prediction of physical and chemical properties

Section 2.4

(CELLO2GO tool)

Sub-cellular localization analysis

Section 2.5

(SOPMA tool)

Prediction of elements of secondary structure i.e. alpha helix, extended strand, beta turn and bends.

Caver Web Server

To predict the catalytic site of protein

Section 2.6

PHYRE2 tool

Three dimensional structure of protein

PyMOL

Visualization of protein structures

Saves server

To predict protein structure quality using Ramachandran plots

Section 2.7

MEME suite

To predict the conserved motifs of proteins

Interproscan

To estimate ontology of predicted conserved motif domains

Section 2.8

Antigenic peptides tool

To detect antigenic determinant sites of proteins

Comment 13

Section 3.1 looks like methodology. The authors are directed to discuss phylogenetic tree. On which basis these strains were selected for phylogenetic analysis.

Reply to reviewers comment

As we are trying to explore the GUS enzyme among the bacteria inhabiting the GIT of normal and breast cancer patients. Therefore, it is important to get insight into the evolution of GUS coding uidA gene of these bacteria. For this purpose, phylogenetic tree has been constructed using GUS enzymes sequences (Figure 1). According to this phylogeny study, F. prausnitzii 1 and S. suis are sharing the same clade so are closely related to each other. E. coli strain K12 and S. enterica are also originating from the same branch point. M. bacterium is not sharing closeness with any of the bacteria as it is not sharing clade. S. xylosus and S. caeli are closely related with each other and also shared clade with S. hemolyticus. These three bacteria are also related with R. intestinalis. P. acnes and E. gallinarum 2 are sharing closeness. S. aquatilis, F. prausnitzii 3 and C. amalonaticus are also related more with each other as compared to other bacteria. C. comes and Bacillus sp. are showing the common ancestry due to origin from common branch point. L. rhamnosus and F. prausnitzii 2 are sharing clade with each other. E. gallinarum 1 is also originating from the same branch point as that of L. rhamnosus and F. prausnitzii 2.

Comment 14

Section 3.3 add figure or table.

Reply to reviewers comment

In section 3.3, Figure and Table are already added. See the Table 2 and Figure 2.

Comment 15

Writing on the yellow color is not visible change the color.

Reply to reviewers comment

The color has been removed. It was added by mistake.

Comment 16

Section 3.7 generic name must be abbreviated, Like L. rhamnosus.

Reply to reviewers comment

The generic name has been abbreviated.

Comment 17

Provide details of figure 7 in section 3.7.

Reply to reviewers comment

Detail is already given in section 3.7. Positions and number of sequences that might be involved in antigenic propensity were different in all proteins. i. e. L. rhamnosus (24), R. intestinalis (27), C. comes (30), F. prausnitzii 1 (25), F. prausnitzii 2 (29), F. prausnitzii 3 (27), P. acnes (22), M. bacterium (23), S. xylosus (19), S. suis (20), Bacillus sp. M4U3P1 (22), E. coli (strain K12) (26), S. aquatilis NBRC 16722 (27), C. amalonaticus (28), E. gallinarum 1 (26), E. gallinarum 2 (27), S. enterica  (27), S. caeli (22, S. haemolyticus (23).

Comment 18

Mostly results are poorly discussed. These can be consulted with the professional such as phylogenetic and motif results.

Reply to reviewers comment

The results have been improved as described in Results sections 3.1 and 3.6.

Comment 19

The authors should provide relation of the GUS enzymes (GUSOME) genes and its associated pathways with the help of related previous studies. Also compare the results of these studies with the current study.

Reply to reviewers comment

The gmGUS play their role in estrobolome by reversing the glucuronidation process and catalyze estrogens activation by breaking glucuronic moiety. Estrogen and estrone glucuronides that might be the substrates for gmGUS include 17-alpha estradiol 17-O, 17-beta estradiol 17-O, 17-beta estradiol 3-O, 2-hydroxy-17-beta estradiol 2-O, 2-hydroxy-17-beta estradiol 3-O, 4-hydroxy 17-beta estradiol-3-O, 4-hydroxy 17-beta estradiol-4-O, estrone 3-O, 2-hydroxy estrone-2-O, 2-hydroxy estrone-3-O, 4-hydroxy estrone-3-O, 16-alpha-17-beta estriol 3-O, 16-beta-17-beta estriol 3-O, 16-alpha-17-alpha estriol 3-O, 16-alpha-17-beta estriol 16-O, 16-alpha-17-alpha estriol 16-O, 16-beta-17-beta estriol 16-O, 16-alpha-17-beta estriol-16-O, 16-alpha-17-alpha estriol-17-O and 16-beta-17-beta estriol-17-O, respectively [30].

This reaction releases aglycones. Process of deconjugation occurs as estrogens after GIT via bile [24]. Estrogens without deconjugation, due to high polarity and hydrophilicity, get dissolved in blood and are removed through kidney as urine. However, on deconjugation estrogens via reabsorption in mucosa enters the portal vein [32]. Due to association of high estrogen concentration with breast cancer gut microbes and breast cancer axis has been as emerging research area.

Comparison of these studies with previous has been given in detail in discussion section (see reply to reviewers comment 20 and 21).

Comment 20

Mainly the discussion is not cited.

Reply to reviewers comment

The discussion has been cited as follows;

The gmGUS play their role in estrobolome by reversing the glucuronidation process and catalyze estrogens activation by breaking glucuronic moiety. Estrogen and estrone glucuronides that might be the substrates for gmGUS include 17-alpha estradiol 17-O, 17-beta estradiol 17-O, 17-beta estradiol 3-O, 2-hydroxy-17-beta estradiol 2-O, 2-hydroxy-17-beta estradiol 3-O, 4-hydroxy 17-beta estradiol-3-O, 4-hydroxy 17-beta estradiol-4-O, estrone 3-O, 2-hydroxy estrone-2-O, 2-hydroxy estrone-3-O, 4-hydroxy estrone-3-O, 16-alpha-17-beta estriol 3-O, 16-beta-17-beta estriol 3-O, 16-alpha-17-alpha estriol 3-O, 16-alpha-17-beta estriol 16-O, 16-alpha-17-alpha estriol 16-O, 16-beta-17-beta estriol 16-O, 16-alpha-17-beta estriol-16-O, 16-alpha-17-alpha estriol-17-O and 16-beta-17-beta estriol-17-O, respectively [30].

This reaction releases aglycones. Process of deconjugation occurs as estrogens after GIT via bile [24]. Estrogens without deconjugation, due to high polarity and hydrophilicity, get dissolved in blood and are removed through kidney as urine. However, on deconjugation estrogens via reabsorption in mucosa enters the portal vein [32]. Due to association of high estrogen concentration with breast cancer gut microbes and breast cancer axis has been as emerging research area.

With reference to sub-cellular localization and physicochemical properties, bacteria in breast cancer patients were found to be more diverse as compared to those reported in normal tissues. The diversity of GUS enzyme found in present study is consistent with the literature [50]. As this protein in Ruminococcus gnavus has been reported to show similarity of 69%, 61%, 59% and 58% with L. gasseri, E. coli, C. perfringens and S. aureus, respectively [51]. So,, this study has further strengthened the diversified nature of GUS enzyme.

The isoelectric point analysis revealed that GUS of S. aquatilis NBRC 16722 was alkaline with pI value 8.92 while in all other cases, GUS showed acidic pI. The pI value for S. caeli is consistent with earlier reported pI value for E. coli HGU-3 [52]. Literature reports that GUS enzyme activity increases at high pH and cause cancer [53]. According to this information, in present study GUS enzyme of S. aquatilis showed alkaline nature as per its pI. While enzymes of all other bacteria except S. caeli exhibited low acidic pI which might have some association with cancer causing potential of this enzyme.

Presence of these large number of antigenic peptides gives a clue about the possible immunomodulatory role of these bacteria. Due to the variety in antigenic peptides, catalytic sites and the adjacent loop structures, anti-cancer medication therapy can be managed via GUS enzyme inhibition also reported in literature [50]. High concentration and deglucuronidation potential of GUS enzyme in breast cancer patients can also be used for bioactivation of glucuronide anticancer prodrugs [61].

Comment 21

The authors are directed to cite the relevant literature.

Reply to reviewers comment

The relevant literature is cited in discussion section as follows;

The gmGUS play their role in estrobolome by reversing the glucuronidation process and catalyze estrogens activation by breaking glucuronic moiety. Estrogen and estrone glucuronides that might be the substrates for gmGUS include 17-alpha estradiol 17-O, 17-beta estradiol 17-O, 17-beta estradiol 3-O, 2-hydroxy-17-beta estradiol 2-O, 2-hydroxy-17-beta estradiol 3-O, 4-hydroxy 17-beta estradiol-3-O, 4-hydroxy 17-beta estradiol-4-O, estrone 3-O, 2-hydroxy estrone-2-O, 2-hydroxy estrone-3-O, 4-hydroxy estrone-3-O, 16-alpha-17-beta estriol 3-O, 16-beta-17-beta estriol 3-O, 16-alpha-17-alpha estriol 3-O, 16-alpha-17-beta estriol 16-O, 16-alpha-17-alpha estriol 16-O, 16-beta-17-beta estriol 16-O, 16-alpha-17-beta estriol-16-O, 16-alpha-17-alpha estriol-17-O and 16-beta-17-beta estriol-17-O, respectively [30].

This reaction releases aglycones. Process of deconjugation occurs as estrogens after GIT via bile [24]. Estrogens without deconjugation, due to high polarity and hydrophilicity, get dissolved in blood and are removed through kidney as urine. However, on deconjugation estrogens via reabsorption in mucosa enters the portal vein [32]. Due to association of high estrogen concentration with breast cancer gut microbes and breast cancer axis has been as emerging research area.

With reference to sub-cellular localization and physicochemical properties, bacteria in breast cancer patients were found to be more diverse as compared to those reported in normal tissues. The diversity of GUS enzyme found in present study is consistent with the literature [50]. As this protein in Ruminococcus gnavus has been reported to show similarity of 69%, 61%, 59% and 58% with L. gasseri, E. coli, C. perfringens and S. aureus, respectively [51]. So,, this study has further strengthened the diversified nature of GUS enzyme.

The isoelectric point analysis revealed that GUS of S. aquatilis NBRC 16722 was alkaline with pI value 8.92 while in all other cases, GUS showed acidic pI. The pI value for S. caeli is consistent with earlier reported pI value for E. coli HGU-3 [52]. Literature reports that GUS enzyme activity increases at high pH and cause cancer [53]. According to this information, in present study GUS enzyme of S. aquatilis showed alkaline nature as per its pI. While enzymes of all other bacteria except S. caeli exhibited low acidic pI which might have some association with cancer causing potential of this enzyme.

Presence of these large number of antigenic peptides gives a clue about the possible immunomodulatory role of these bacteria. Due to the variety in antigenic peptides, catalytic sites and the adjacent loop structures, anti-cancer medication therapy can be managed via GUS enzyme inhibition also reported in literature [50]. High concentration and deglucuronidation potential of GUS enzyme in breast cancer patients can also be used for bioactivation of glucuronide anticancer prodrugs [61].

Comment 22

In conclusion also recommend techniques which can be helpful for the manipulation of these enzymes and its related genes.  

Reply to reviewers comment

The modifications and manipulation strategy has been mentioned in conclusion section as follows;

Modifications may include alteration in structure of active sites participating in deglucuronidation by inducing mutations at specific points in uidA gene and deletion of conserved protein motifs. For these modifications site directed mutagenesis can be performed which may lead to destabilization of protein thus inhibiting its estrogen reactivation potential.

  1. Urbaniak, C.; Gloor, G.B.; Brackstone, M.; Scott, L.; Tangney, M.; Reid, G. The microbiota of breast tissue and its association with breast cancer. Applied and environmental microbiology 2016, 82, 5039-5048.
  2. He, C.; Liu, Y.; Ye, S.; Yin, S.; Gu, J. Changes of intestinal microflora of breast cancer in premenopausal women. European Journal of Clinical Microbiology & Infectious Diseases 2021, 40, 503-513.
  3. Costantini, L.; Magno, S.; Albanese, D.; Donati, C.; Molinari, R.; Filippone, A.; Masetti, R.; Merendino, N. Characterization of human breast tissue microbiota from core needle biopsies through the analysis of multi hypervariable 16S-rRNA gene regions. Scientific reports 2018, 8, 1-9.
  4. Geourjon, C.; Deleage, G. SOPMA: significant improvements in protein secondary structure prediction by consensus prediction from multiple alignments. Bioinformatics 1995, 11, 681-684.
  5. Stourac, J.; Vavra, O.; Kokkonen, P.; Filipovic, J.; Pinto, G.; Brezovsky, J.; Damborsky, J.; Bednar, D. Caver Web 1.0: identification of tunnels and channels in proteins and analysis of ligand transport. Nucleic acids research 2019, 47, W414-W422.
  6. Consortium, U. UniProt: a hub for protein information. Nucleic acids research 2015, 43, D204-D212.
  7. Sievers, F.; Higgins, D.G. The clustal omega multiple alignment package. In Multiple sequence alignment; Springer: 2021; pp. 3-16.
  8. Kumar, S.; Stecher, G.; Tamura, K. MEGA7: molecular evolutionary genetics analysis version 7.0 for bigger datasets. Molecular biology and evolution 2016, 33, 1870-1874.
  9. Saitou, N.; Nei, M. The neighbor-joining method: a new method for reconstructing phylogenetic trees. Molecular biology and evolution 1987, 4, 406-425.
  10. Zuckerkandl, E.; Pauling, L. Evolutionary divergence and convergence in proteins. In Evolving genes and proteins; Elsevier: 1965; pp. 97-166.
  11. Khan, S.; Vihinen, M. Spectrum of disease-causing mutations in protein secondary structures. BMC Structural Biology 2007, 7, 1-18.

Reviewer 2 Report

The article present computatational characterization of estrogen reactivating β-glucuronidase enzyme in GIT associated microbiota of normal human and breast cancer patients which is well designed and analyzed. However experimental framework is missing in this work but related to bioinformatics. The findings may help in further experimental work on the topic.

In abstract main results should be added.

Also add future recommendations in the last paragraph of the introduction

In methodology most of the time methods are not cited and also not fully explained.

Some results sections looks like methods these must be imroved.

In introduction add mechanism of the gene expression of respective enzymes

The authors should focus on the typos and grammatical mistakes

Author Response

Rebutal Letter

Reviewers comment 1

In abstract main results should be added.

Reply to reviewers comment

The results have been added in abstract in detail (See abstract).

Reviewers comment 2

Also add future recommendations in the last paragraph of the introduction

Reply to reviewers comment

Future recommendations have been added in last paragraph (see introduction section).

Reviewers comment 3

In methodology most of the time methods are not cited and also not fully explained.

Reply to reviewers comment

The methods have been fully explained and cited (see methodology section).

Reviewers comment 4

Some results sections looks like methods these must be improved.

Reply to reviewers comment

The results sections has been improved (see results section).

Reviewers comment 5

In introduction add mechanism of the gene expression of respective enzymes

Reply to reviewers comment

The mechanism of gene expression of GUS enzyme has been incorporated in introduction section in paragraph 5 and Figure 1.

Reviewers comment 6

The authors should focus on the typos and grammatical mistakes

Reply to reviewers comment

The grammatical mistakes and typos have been improved.

Round 2

Reviewer 1 Report

The suggested changes are made by authors but there are still issues with English language.